# Simultaneous Quantification of 16 Bisphenol Analogues in Food Matrices

**DOI:** 10.3390/toxics11080665

**Published:** 2023-08-02

**Authors:** Fiorella Lucarini, Rocco Gasco, Davide Staedler

**Affiliations:** 1Department of Biomedical Sciences, University of Lausanne, 1011 Lausanne, Switzerland; 2School of Engineering and Architecture, Institute of Chemical Technology, University of Applied Sciences and Arts of Western Switzerland, 1700 Fribourg, Switzerland; 3Department for Environmental and Aquatic Sciences, University of Geneva, 1211 Geneva, Switzerland

**Keywords:** bisphenol analogues, canned foods, food matrices, QuEChERS, GC-MS, LC-MS/MS, endocrine disruptors

## Abstract

Exposure to bisphenol analogues can occur in several ways throughout the food production chain, with their presence at higher concentrations representing a risk to human health. This study aimed to develop effective analytical methods to simultaneously quantify BPA and fifteen bisphenol analogues (i.e., bisphenol AF, bisphenol AP, bisphenol B, bisphenol BP, bisphenol C, bisphenol E, bisphenol F, bisphenol G, bisphenol M, bisphenol P, bisphenol PH, bisphenol S, bisphenol Z, bisphenol TMC, and tetramethyl bisphenol F) present in canned foods and beverages. Samples of foods and beverages available in the Swiss and EU markets (*n* = 22), including canned pineapples, ravioli, and beer, were prepared and analyzed using QuEChERS GC-MS. The quantification method was compared to a QuEChERS LC-MS/MS analysis. This allowed for the selective and efficient simultaneous quantitative analysis of bisphenol analogues. Quantities of these analogues were present in 20 of the 22 samples tested, with the most frequent analytes at higher concentrations: BPA and BPS were discovered in 78% and 48% of cases, respectively. The study demonstrates the robustness of QuEChERS GC-MS for determining low quantities of bisphenol analogues in canned foods. However, further studies are necessary to achieve full knowledge of the extent of bisphenol contamination in the food production chain and its associated toxicity.

## 1. Introduction

Bisphenol A (BPA) and its analogues are phenol-based chemicals broadly used in the plastic industry [1,2,3]. With a global production estimated at 5.5 billion tons in 2021 [4], BPA is one of the most commonly produced compounds among bisphenols (BPs). It is widely used as a raw material for the synthesis of polycarbonates and epoxy resins or as an additive (such as an antioxidant or a stabilizing agent) to improve the properties of plastic materials [1]. Many products used in daily life contain BPA, including water pipes, food containers, medical equipment, toys, and electronics. Globally, it has been demonstrated that populations have been chronically exposed to BPA via different pathways, including oral, dermal, and hand-to-mouth transfer, as well as other mechanisms [4,5]. However, because of its adverse impact on human health and the environment, the contamination of commercial products with BPA has recently attracted tremendous attention. BPA is considered an endocrine-disrupting chemical (EDC), responsible for impeding the function of estrogenic and androgenic hormones [6,7,8,9,10,11]. In addition, it is a highly polluting substance, mainly generated by effluents from the plastic industry, and is commonly found among the contaminants identified in soils and waters [12,13]. The use of BPA has therefore been limited or prohibited in many countries [14,15] leading to an increasing demand by the plastic industry for bisphenol analogues to replace BPA.

Several bisphenol analogues have been produced to replace BPA in research and industry (Figure 1). Bisphenols S, F, and AF (BPS, BPF, and BPAF), which display similar chemical structures in comparison to BPA, are the most commonly used substituents in industry [16,17,18,19]. They exhibit similar stability and thermoplastic properties, but with different reactivities and processabilities, which sometimes hinder their efficient use in the manufacture of polymers. Although studies about the toxicological behavior of the bisphenol analogues are limited, several reports demonstrate a wide variety of different toxicological mechanisms, including endocrine disruptive effects, cytotoxicity, genotoxicity, or neurotoxic effects [15,17,20,21]. Despite the potential risks to human health, no restrictions have been implemented for most of the BPs on the market [20,22]. Therefore, the development of rapid and low-cost techniques for the determination of BPA analogues with high sensitivity is urgently needed. The detection and quantification of bisphenols in food matrices have already been described in the literature and are usually performed using liquid chromatography or gas chromatography coupled to mass spectrometry (LC-MS and GC-MS) or tandem mass spectrometry (LC-MS/MS and GC-MS/MS) [23]. However, the reported methodologies are usually limited to specific bisphenol analogues and cannot be applied to a broad range of BPs because they are generally optimized and developed specifically for a limited number of compounds; furthermore, only a limited number of studies investigating the presence of bisphenols in food and beverages are reported. Other studies have employed aptasensors to detect BPA in food matrices and canned foods [23,24,25,26,27,28]. However, long turnaround times, expensive equipment, and laborious processing prevent their widespread diagnostic use. Therefore, the development of new analytical methods allowing for the detection of a high number of different bisphenols in food is needed.

There are several approaches for the extraction of bisphenols from foods and materials. These are based on different simulants, generally on an aqueous basis, while extraction is carried out by solid-liquid/liquid-liquid extraction techniques, or by solid-phase extraction (SPE) and the QuEChERS (Quick, Easy, Cheap, Effective, Rugged, and Safe) technique [29,30,31]. Recently, some innovative techniques related to SPE have emerged, such as the use of molecularly imprinted polymers as well as techniques based on micro-extraction [32,33]. Among these techniques, classical SPE and QuEChERS are among the most widely used in quality control and surveillance analysis laboratories and have shown great robustness in terms of reliability and matrix effect reduction [34,35]. In addition, these methods are simple to implement, easily accessible commercially, and also allow for a reduction in the use of environmentally harmful solvents. Therefore, in this work, it was decided to focus on the optimization of the QuEChERS technique for the analysis of a large number of bisphenols, which allows not only to obtain reliable results but also to reduce the number of consumables used in the laboratory.

In this work, we developed and compared a new extraction and analytical method for selective and efficient detection of bisphenol analogues in canned food and beverages. The QuEChERS technique was used in combination with GC-MS for the simultaneous quantification of 16 bisphenol analogues (bisphenol A (BPA), bisphenol AF (BPAF), bisphenol AP (BPAP), bisphenol B (BPB), bisphenol BP (BPBP), bisphenol C (BPC), bisphenol E (BPE), bisphenol F (BPF), bisphenol G (BPG), bisphenol M (BPM), bisphenol PH (BPPH), bisphenol P (BPP), bisphenol S (BPS), bisphenol TMC (BPTMC), bisphenol Z (BPZ), and tetramethyl bisphenol F (TMBPF)), in various canned foods and beverages. Bisphenol analogues can also be found in non-canned foods and pose a clear risk to human health. However, due to the greater concentrations of BPs in canned foods [36], it is imperative to have a rapid detection method for several analogues simultaneously so that daily intake can be monitored and controlled. Therefore, the detection of BPs in canned food remains an area of great interest. Several studies have highlighted the toxicity profiles of bisphenol analogues as well as their interactions with other compounds that can lead to increased bioavailability and uptake of BPA in cells [6,22,36,37,38,39,40,41,42,43,44,45,46,47]. However, despite the established and growing evidence of their harmful effects on human health, BPA and its analogues are not prohibited, with the exception of their use in cosmetic substances [48] and in plastic infant feeding bottles [49]. In recent years, European Union regulations have focused on detecting the limits for BPA products across a range of sectors [50,51,52,53,54,55,56]. In the case of plastic infant feeding bottles, BPA is prohibited according to EU regulation [49]. In plastic materials and articles intended to come into contact with food, the migration limit is 0.6 mg/kg [53], in varnishes and coatings intended to come into contact with food, the limit is 0.05 mg/kg [51], and for toys intended for children, the limit is 0.04 mg/l [55]. However, even if the migration of BPA and its analogues is restricted within the 0.05 mg/kg limit in canned foods, smaller concentrations of these chemicals are still ingested and detectable in the human body. This demonstrates the need for novel analytical methods for the simultaneous analysis of a wide variety of bisphenols.

## 2. Materials and Methods

### 2.1. Chemicals

Bisphenol A (≥99%), bisphenol AF (≥99%), bisphenol AP (≥99%), bisphenol B (≥98%), bisphenol BP (≥98%), bisphenol C (≥98%), bisphenol E (≥98%), bisphenol F (≥98%), bisphenol G (≥98%), bisphenol M (≥99%), bisphenol P (≥99%), bisphenol PH (≥99%), bisphenol S (≥98%), bisphenol TMC (≥97%), and bisphenol Z (≥99%) were purchased from Neochema Gmbh pre-dissolved in acetonitrile and in a concentration of 100 ppm (µg/mL) (Stock solution 1: standards (STDs) mix 16 Bisphenols). This stock-standard mixture was stored at −20 °C. The tetramethyl bisphenol F (≥99%) standard was purchased from Sigma Aldrich Chemie GmbH (Taufkirchen, Germany). The stock solution was made by weighing 10 mg of standard and dissolving it in 10 mL of methanol. The internal standard (ISTD), bisphenol A d-16, was purchased from Neochema Gmbh (Bodenheim, Germany) pre-dissolved in acetonitrile with a concentration of 100 ppm (µg/mL). Working solutions for calibrations were prepared by dilution of the stock standard mixture. The internal standard working solution was prepared separately by diluting the stock standard with acetonitrile (Sigma-Aldrich).

A solution of Bis(trimethylsilyl)trifluoroacetamide (BSTFA) was used for BPs derivatization and was purchased from Sigma Aldrich. Magnesium sulfate anhydrous (≥99.5%), sodium chloride, dichloromethane (≥99.8%), methanol (≥99.9%), and acetonitrile (≥99.9%) were also purchased from Sigma Aldrich. Nanopure water was provided by an ultrapure water system (ariumPro, Sartorius, Göttingen, Germany). The solid phase extraction method was carried out with a CHROMABOND^®^.HLB cartridge (3 mL, 200 mg), which was purchased from Macherey-Nagel, Düren, Germany. The QuEChERS clean-up was performed with CHROMABOND^®^ Mix XX (1.20 g MgSO_4_, 0.40 g CHROMABOND^®^ Diamino), which was purchased from Macherey-Nagel.

### 2.2. Samples and Sample Preparation

A total of 22 samples of canned food and beverages, which were all readily available in Lausanne and the Swiss market, were analyzed. All samples were stored at room temperature prior to preparation and returned to the refrigerator/freezer once preparation was complete. Food samples comprised canned pineapple (*n* = 2), canned peaches (*n* = 1), canned ravioli (*n* = 5), farce vol-au-vent (*n* = 2), soup (*n* = 2), fruit puree (*n* = 5), canned tuna (*n* = 1), cola light (*n* = 1), lemon (*n* = 1), and beer (*n* = 2). Canned food samples were homogenized with an electric blender. In the case of the canned fruit sample, the solid and liquid parts were analyzed separately to study the migration effects of bisphenols. 10 g of each homogenized or liquid sample were added to a 50-milliliter tube, followed by 100 µL of ISTD 1 ppm. The solid sample and any samples that were in the liquid phase once homogenized had 5 mL of Evian water added to facilitate liquid-liquid extraction.

### 2.3. Extraction Method

Ten milliliters of acetonitrile was added to each of the samples, followed by further homogenization with the vortex. Magnesium sulfate anhydrous (MgSO_4_) and sodium chloride (NaCl) were added (4 g and 1 g, respectively) to perform salting out of the liquid-liquid extraction phase of the QuEChERS method. The sample was then shaken either by hand or in the vortex for 1 min. The samples were then added to a centrifuge at 1000 rpm for 15 min. An amount of 5 mL of supernatant in the organic phase was collected and transferred into the CHROMABOND^®^ Mix XX and further shaken either by hand or vortex. The sample was then filtered using a PTFE 0.45-micrometer filter into a 40-milliliter glass tube. The filtered solution was then evaporated under N_2_ flow.

### 2.4. Derivatization

Derivatization of BPs was performed with bis(trimethylsilyl)trifluoroacetamide (BSTFA) at a temperature of 60 °C for 45 min. Four different volumes of BSTFA were tested: 20, 50, 70, and 100 μL. For the derivatization tests, 500 μL of STD mix at 1 ppm were put in a vial and evaporated. The different volumes of BSTFA were added, and for each volume, acetonitrile was added to reach a total volume of 100 µL. Following derivatization and cooling at room temperature, the samples were subjected to GC-MS analysis.

### 2.5. GC-MS Parameters

Following extraction using the QuEChERS method, the samples were subjected to gas chromatography-mass spectrometry model GCMS-QP 2010 Ultra (Shimadzu Corporation, Kyoto, Japan) in EI mode and utilizing LabSolutions software. The instrument was equipped with an OPTIMA-5 MS column with a diameter of 25 mm, a length of 30 m, and a film thickness of 0.25 µm. Helium was utilized as the carrier gas at a constant pressure of 58.4 kPa with a flow rate at an initial temperature of 5.28 mL/min, a total flow of 24.4 mL/min, and a column flow of 1.09 mL/min. The ion source temperature was adjusted to 250 °C, while the interface temperature was 280 °C. A selected ion monitoring (SIM) mode was then used to quantify the analytes in the samples and the standard mixture of the calibration curve, as shown in Table 1 [52].

### 2.6. Calibration Curve and Controls for GC-MS

The calibration curve samples were prepared in water and spiked with 100 µL of ISTD, 1 ppm, and with various volumes of a standard mixture of 16 BPs, 1 ppm, and a standard mixture of 16 BPs, 100 ppb, to produce concentrations of 50, 10, 5, 1, 0.5, and 0.1 µg/L (ppb). Blank samples were prepared in the same way as the calibration curve with 100 µL of ISTD, 1 ppm. The calibration curve and the blank followed the same preparation procedure as the samples. Each standard sample was prepared in duplicate and analyzed. Recovery tests (i.e., samples prepared by adding a known amount of standards and extracted with the same method described above) and triplicate measurements were conducted to study the efficiency and reliability of the extraction method. The limit of detection (LOD) and limit of quantification (LOQ) were calculated for each bisphenol analyzed according to Equations (1) and (2), respectively.
(1)$$LOD=3\;\frac{S}{b}$$
(2)$$LOQ=10\;\frac{S}{b}$$
where *S* is the residual standard deviation and *b* is the slope of the calibration curve.

### 2.7. LC-MS/MS Parameters

Similarly, for the QuEChERS GC-MS (SIM) method, the sample preparation and extraction, as well as the evaporation and recovery of the extract, were carried out under the same conditions as for QuEChERS LC-MS/MS. No sample derivatization was performed, but after evaporation, the crude was resuspended in 1 mL of water and directly analyzed by LC-MS/MS.

Chromatographic analysis was performed using a Shimadzu LCMS-8060. This was coupled to a triple quadrupole and equipped with three solvent modules: Nexera X2 LC-30AD, an autosampler Nexera X2 SIL-30AC, a column over unit CTO-20AC, two degassing units GDGU-205R, and a valve unit FCV-20AH_2_. All instruments were Shimadzu, Japan, models. The LC separation was conducted on an Agilent Poroshell 120 EC-C18-treated column (Agilent technologies, Santa Clara, CA, USA; 2.7 μm, 4.6 mm × 50 mm). The oven temperature was set and maintained at 40 °C, with a temperature limit of 90 °C. The injection volume was 20 μL. The flow rate was 0.25 mL/min, and the total data acquisition time was 30 min. The mobile phase consisted of a mixture of water, Nanopure 5 mM ammonium acetate (solvent A), and methanol (solvent B). The analysis was conducted in a gradient mode where the organic mobile phase, solvent B, increased linearly. The time and percentages were as follows: initially 25% for 3 min, then increased to 100% over 3 to 20 min and maintained at 100% during 20 to 24 min; at 24 min, the eluent was restored to the initial conditions for 6 min to equilibrate the column for the next injection. The pressure limit was 0 to 1000 bar. The mass spectrometry analysis was carried out on a triple quadrupole with an electrospray (ESI) source operating in negative mode. The interface temperature was set at 400 °C, and Argon was utilized as a carrier gas. The nebulizing gas flow was 2 L/min, the heating gas flow was 10 L/min, and the drying gas flow was 10 L/min. The desolvation temperature was 650 °C. MS data were acquired in the 100–1000 m/z range. Data acquisitions were performed using LabSolutions LCX3-TQ8060, while data processing was performed with LabSolutions Insight. All parameters related to the analytes analyzed are shown in Table 2.

### 2.8. Calibration Curve and Controls for LC-MS/MS

The calibration curve samples were prepared in the same manner as the QuEChERS GC-MS (SIM) Method of Extraction to produce concentrations of 50, 10, 5, 1, 0.5, and 0.1 µg/L (ppb). Blanks, recoveries, and triplicates were also performed as described in Section 2.6.

### 2.9. Statistical Analysis

The statistical treatment of the data set obtained from the analysis follows the “Guidelines for performance criteria and validation procedures of analytical methods used in controls of food contact materials—EUR 24105 EN (2009)” [57]. All the statistical values were calculated with the regression analysis output calculated with Excel software, Office 365 version (Microsoft, Redmond, WA, USA).

## 3. Results and Discussion

### 3.1. Optimization of the Derivatization

The chromatographic response of bisphenol analogues was determined through derivatization due to their high polarity and non-volatile nature using silylation and performed with N, O-bis(trimethylsilyl)trifluoroacetamide (BSTFA; IUPAC: trimethylsilyl 2,2,2-trifluoro-N-trimethylsilylethanimidate) [52]. The results of the derivatization tests (Figure 2) showed that the best results are linked to the use of 50 μL of BSTFA/50 μL of acetonitrile and 70 μL of BSTFA/30 μL of acetonitrile. For this reason, 50 μL of BSTFA and 50 μL of acetonitrile were chosen for the derivatization of the samples.

### 3.2. Statistical Validation of the Extraction and Analytical Method

The developed method employs an internal standard (BPA-d16), which enables it to account for variation in the response of the chromatographic system, the exact volume of sample injected, and all losses during the extraction method. Calibration curves were then constructed using the ratio of C_BPs_/C_BPA-d16_. Table 3 shows the details of the extraction method, demonstrating a good correlation coefficient of determination (R^2^) and instrumental trueness (C_found_/C_nominal_ × 100) and low LOQ and LOD for each bisphenol. Moreover, the linearity was evaluated with two different methods: residues for different calibration levels and the adequateness of the linearity model. By residue calculation, results were obtained that confirm the linearity of the method for every analyte. The data produced by the adequateness of the linearity model results confirms the predictive ability of each calibration curve [57].

The selectivity of the method was evaluated by the analysis of different blanks, showing high selectivity for each analyte and no interferences. Moreover, at least one blank was analyzed every time the analysis was performed.

A short-term repeatability test was conducted and statistically treated according to UNI ISO 5725-1. The repeatability of the test was confirmed using the Horwitz equation [58].

Evaluation of the recovery of the QuEChERS extraction methods was performed on every sample for a standard concentration of 10 ppb. Table 4 shows the results of the recovery for each type of sample obtained as geometric means of the recoveries of the single samples analyzed.

For the range of concentrations analyzed, the UNI CEI EN ISO/IEC 17025:2005 indicates a different acceptable mean recovery (%) range. For lower concentrations, this range increases: in the order of 10^−19^ (10 µg/kg), the range of mean recovery is 60–115% [59]. As reported in Table 4, in the fruit, vegetable, beverage, and canned food matrices, all recoveries were within acceptable limits. The complexities of the matrices within canned food, however, presented criticalities, especially in the case of BPS, where some recoveries were outside of the acceptable range. An exception was found in the tuna matrix with BPAP, BPM, BPP, BPBP, and BPPH, where recoveries were in the range of 25–46%.

### 3.3. Quantitative Results

In only 2 samples (canned pineapple and fruit puree) of the 22 analyzed, no bisphenols were detected. Of the sixteen possible analytes analyzed, the samples were positive for eight of the bisphenols: BPAF, BPF, BPE, BPA, BPB, BPS, BPM, and TMBPF (Table 5). The most frequently detected analytes, and indeed those found at higher concentrations, were BPA and BPS. BPA was detected in 78% of all cases in concentrations between 3.21–40.65 µg/kg, while BPS was found in 48% of cases in the concentration range of 5.58–11.11 µg/kg.

In the vegetable soup and canned fruit (Table 5A), low concentrations of BPAF, BPF, BPE, BPA, and BPB were found, with 3.21 µg/kg of BPA being detected. In three of the five fruit puree samples analyzed (Table 5B), a high concentration of BPS was found along with lower concentrations of BPE and BPB. In one of the samples, BPA was detected at 12.72 µg/kg.

In complex food matrices such as ravioli (Table 5C), all samples contained similar concentrations of BPAF, BPA, BPB, and BPS, with a similar pattern being shown with the vol-au-vents. With the tuna sample, analysis revealed the presence of BPE, BPA, BPS, and BPM.

In beverages (Table 5D), only BPA, TMBPF, and BPS were detected. BPA and BPS were found at concentrations below the LOQ. Only TMBPF was found at concentrations above the LOQ in the two beer samples.

In order to test the robustness of the QuEChERS extraction method, a confirming test was conducted, changing the instrument and method of analysis and involving the use of the LC-MS/MS. The samples that showed higher positive concentrations, especially of BPA, were the ones re-analyzed with the LC-MS/MS. Two types of different ravioli and the farce for vol-au-vent were subjected to analysis with GC-MS and LC-MS/MS. The goal is to compare the results obtained with the GC-MS analysis, specifically the BPA concentration in the samples that resulted in positive results. Samples that showed higher positive concentrations of BPs, especially BPA, were reanalyzed using this method. Ravioli and vol-au-vents were subjected to analysis with both GC-MS and LC-MS/MS with the aim of comparing results (Table 6).

The data obtained from LC-MS/MS for the ravioli was comparable to that found using GC-MS. However, for the vol-au-vents, the BPA concentration deviated, with a concentration of 27.35 µg/kg using LC-MS/MS and 37.47 µg/kg with GC/MS analysis.

The small difference between the values obtained for GC-MS and LC-MS/MS confirms that the use of GC after QuEChERS extraction is also a valid method of analysis of bisphenols in complex matrices.

When compared with the migration limit set out by the EU and Switzerland for food in contact with coated cans, all samples were in accordance with the regulations, with BPs present at concentrations <50 µg/kg. Results for the concentrations of BPA found in canned pineapple concurred with results of a similar study conducted by Cunha et al., with concentration ranges between 4 and 10 μg/kg [60].

The fruit puree samples were not collected from a can yet, but high concentrations of BPs, in particular BPS, were found in the range of 6.65–11.11 µg/kg, with one sample revealing BPA at 12.72 µg/kg. Migration from the epoxy resin was not possible in these samples, lending weight to the theory of contamination during the food production chain [36].

BPA was also found in ravioli samples in the range of 21.67–36.80 µg/kg and from 37.47–40.65 µg/kg in the vol-au-vents. A 2010 study focused on the presence of BPA in the Belgian market, finding a concentration of 73.1 µg/kg for ravioli and 29.3 µg/kg for cream of chicken soup [61], a sample that is chemically comparable to the vol-au-vents analyzed. A concentration of 10.82 µg/kg of BPA was found in the tuna, which was comparable with a study of both canned and non-canned tuna, which had concentrations in the range of 1.0–99.9 µg/kg [62]. Based on the results obtained, it appears that the presence of fat or a matrix increased the migration and retention of bisphenols inside foods. Interestingly, a lower concentration of bisphenols was found in canned fruit and vegetable soup than in food matrices such as ravioli, vol-au-vent, and tuna, which could be linked to the high octanol-water partition coefficient presented by most of the bisphenol analogues.

## 4. Conclusions

The aim of this study was to develop an extraction and analytical method suitable for the detection of bisphenol analogues in complex food matrices. We have successfully demonstrated the acceptability of QuEChERS-GC-MS, also compared to QuEChERS LC-MS/MS. Indeed, the use of GC-MS over LC-MS/MS offers a number of advantages, such as the fact that GC-MS is a much less expensive instrument than LC-MS/MS, is more widely used, and does not require the use of solvents or produce waste solvents. This study shows that the QuEChERS technique is a viable choice for the extraction of BPs from food matrices. QuEChERS extraction allows not only an efficient extraction with a satisfactory recovery, but it is also sustainable from a sustainability perspective, allowing to reduce the amount of solvents and consumables used in the laboratory.

An important point to note is the number of bisphenol analogues included in this study. Sixteen in total were investigated with the method developed, allowing for simultaneous detection of the concentration of these compounds in canned foods. The results obtained from the samples highlighted the presence of a variety of bisphenol analogues in canned foods, not only BPA. The BPS, BPAF, BPE, BPF, and even TMBPF found in canned foods may be due to the replacement of BPA with these analogues. However, they may also be present due to contamination in the food production chain.

Indeed, the origins of BP contamination in canned foods and beverages are difficult to identify. They can come either from the cans themselves, as part of the formulation of epoxy coatings, or from outside, such as from inks and packaging [63]. In this case, contamination occurs at the time of opening the packages. Thus, contamination can be produced at any stage of the food production chain, and often the same industries that produce BPA products also make use of other bisphenols. More detailed studies are needed to investigate the presence of bisphenol analogues in the canned food industry, not only in the final stages of production of the finished product but throughout the food production chain, with a focus on European and Swiss markets. This will provide a more complete picture of exposure to these compounds and may provide a boost to toxicity studies on this still understudied class of compounds [64].

## Figures and Tables

**Figure 1 toxics-11-00665-f001:**
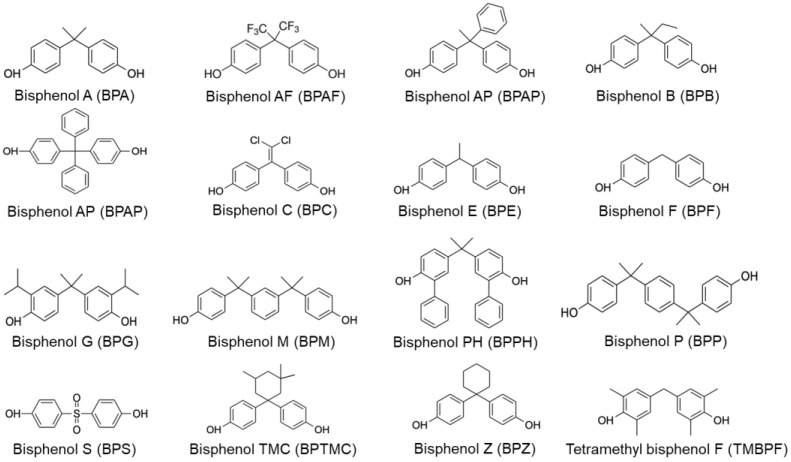
Bisphenol A and associated analogues.

**Figure 2 toxics-11-00665-f002:**
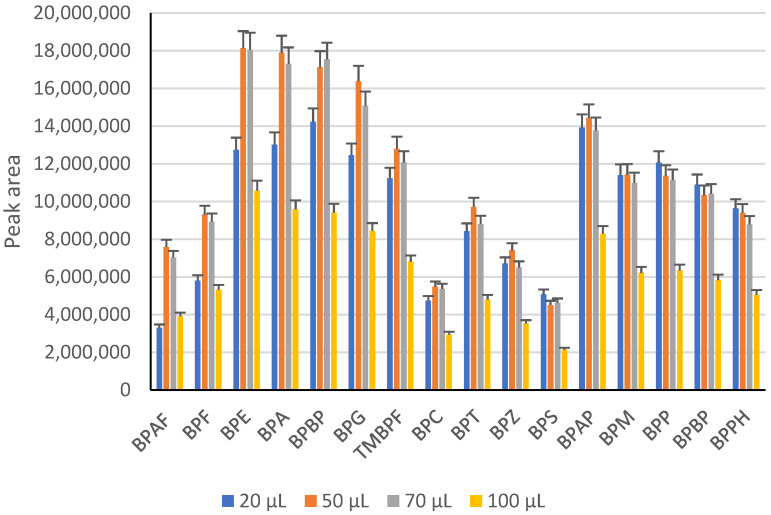
The peaks area (GC-MS (SIM)) of target bisphenols obtained by adding different volumes of BSTFA (blue: 20 μL, orange: 50 μL, grey: 70 μL, and yellow: 100 μL) for the derivatization via silylation.

**Table 1 toxics-11-00665-t001:** Target ion, retention time, and reference ion selection for each analyte during the GC-MS analysis [52].

Name	Ret. Time (min)	Target Ion (m/z)	Ref. Ions (m/z)
BPAF	18.95	411.00	480.00, 412.00, 225.00
BPF	20.285	344.00	179.00, 345.00, 157.00
BPE	20.550	343.00	344.00, 358.00, 193.00
BPA d16	20.790	368.00	369.00, 386.00, 217.00
BPA	20.860	357.00	358.00, 372.00, 207.00
BPB	21.555	357.00	358.00, 191.00, 221.00
BPG	21.790	441.00	442.00, 456.00, 249.00
TMBPF	22.925	385.00	400.00, 386.00, 207.00
BPC	22.970	424.00	426.00, 374.00, 354.00
BPZ	23.950	369.00	412.00, 370.00, 203.00
Bisphenol TMC	24.210	383.00	384.00, 454.00, 397.00
BPS	24.590	394.00	379.00, 135.00, 229.00
BPAP	24.845	419.00	420.00, 269.00, 434.00
BPM	26.125	475.00	476.00, 490.00, 387.00
BPP	27.400	475.00	476.00, 490.00, 230.00
BPBP	28.225	419.00	420.00 331.00, 496.00
BPPH	28.395	509.00	510.00, 542.00, 267.00

**Table 2 toxics-11-00665-t002:** Retention time, precursor ion, and product ion related to every analyte during the LC-MS/MS analysis.

Name	Ret. Time (min)	Acquisition Segment (min)	Precursor Ion (m/z)	Product Ion (m/z)	Collision Energy (eV)
BPS	7.381	3.82–8.32	249.2	108.1 156.0	15 25
BPF	11.469	5.00–12.878	199.1	93.0 105.0	22 22
BPE	12.578	7.00–16.00	213.0	197.9 118.8	12 22
BPA d-16	13.378	9.915	241.1	222.9 141.9	19 24
BPAF	15.332	11.925–16.925	335.0	196.9 176.8	38 45
BPA	13.516	10.067–15.027	226.9	211.8 132.9	19 26
BPB	14.705	11.28–16.28	241.0	211.9 225.9	18 18
BPC	15.021	11.589–16.589	279.0	35.0 71.0	17 16
BPAP	15.435	12.068–17.068	289.0	274.0 195.0	21 26
BPZ	16.041	12.63–17.63	267.2	173.0 222.9	20 25
BPG	17.984	14.754–19.754	311.2	295.1 175.1	29 25
Bisphenol TMC	18.172	14.866–19.866	309.1	237.0 200.0	33 30
BPBP	16.989	13.665–18.665	351.1	273.2 258.0	27 26
Bisphenol M + P	18.037	14.728–19.728	345.0	330.0 133.0	25 35

**Table 3 toxics-11-00665-t003:** QuEChERS GC-MS (SIM) calibration parameters. R^2^: coefficient of determination. Instrumental trueness: (C_found_/C_nominal_ × 100%). LOD: limit of detection (μg/L). LOQ: limit of quantification (μg/L).

Analyte	R^2^	Instrumental Trueness (%)	LOD	LOQ
BPAF	0.999841	95–113	0.39	1.32
BPF	0.999998	99–105	0.04	0.15
BPE	0.999935	97–123	0.25	0.84
BPA	0.999732	94–118	0.51	1.71
BPB	0.999990	87–102	0.10	0.33
BPG	0.999861	96–128	0.37	1.23
TMBPF	0.999923	97–126	0.27	0.92
BPC	0.999573	93–122	0.65	2.16
BP-TMC	0.999658	93–119	0.58	1.93
BPZ	0.999960	89–109	0.20	0.67
BPS	0.998243	86–105	1.66	5.55
BPAP	0.999649	93–145	0.59	1.96
BPM	0.999750	94–112	0.5	1.65
BPP	0.999804	95–107	0.04	0.15
BPBP	0.999913	96–104	0.03	0.10
BPPH	0.998776	74–141	0.11	0.37

**Table 4 toxics-11-00665-t004:** Recovery in % for canned pineapple (*n* = 2), canned peaches (*n* = 1), canned ravioli (*n* = 5), farce vol-au-vent (*n* = 2), soup (*n* = 2), fruit puree (*n* = 5), canned tuna (*n* = 1), cola light (*n* = 1), lemon (*n* = 1), and beer (*n* = 2).

Analyte	Pineapple	Peaches	Soup	Fruit Puree	Ravioli	Farce Vol-Aux-Vent	Tuna	Lemon	Cola Light	Beer
BPAF	91	73	82	79	84	99	100	97	94	103
BPF	81	87	79	75	97	80	82	91	96	100
BPE	91	98	91	84	85	85	95	100	88	99
BPA	97	108	98	92	84	78	94	100	106	99
BPB	68	115	97	90	100	70	91	101	96	101
BPG	91	92	110	104	84	92	51	87	91	92
TMBPF	108	118	119	134	98	102	68	100	101	101
BPC	103	94	96	88	102	97	71	89	93	103
BPTMC	110	120	90	82	93	89	123	86	89	95
BPZ	100	101	102	94	105	92	72	92	100	103
BPS	114	78	105	102	137	156	123	81	106	78
BPAP	100	111	107	97	103	133	46	90	99	94
BPM	108	126	104	108	96	83	29	93	100	111
BPP	107	105	97	90	93	92	28	97	102	96
BPBP	101	107	85	81	92	87	28	97	93	106
BPPH	103	103	107	101	102	104	25	99	95	111

**Table 5 toxics-11-00665-t005:** Quantitative results obtained from analyses. All the values are expressed in μg/kg. All bisphenols were analyzed; for clarity, only bisphenols that have been detected are shown in the tables. (**A**) Fruits and vegetable soup: pineapple pulp and canned water (CW), peaches pulp and canned water (CW), and vegetable soup. (**B**) Fruit purees for children. (**C**) Complex matrices of canned food: ravioli (Rav.), farce vol-aux-vent (Far.), and canned tuna. (**D**) Canned beverages: cola light, lemon soft drink, and beers.

(A)											
Analyte	Pineapple-CW		Pineapple Pulp		Peaches-CW	Peaches-Pulp		Soup (1)	Soup (2)	LOD	LOQ
BPAF	1.78		<LOD		<LOD	<LOQ		<LOQ	<LOD	0.39	1.32
BPF	<LOD		0.18		0.42	0.89		<LOD	<LOD	0.04	0.15
BPE	<LOD		<LOD		<LOQ	2.62		1.24	<LOD	0.25	0.84
BPA	<LOQ		3.21		<LOD	<LOQ		<LOD	<LOQ	0.51	1.71
BPB	<LOD		0.60		<LOD	<LOD		<LOD	<LOD	0.10	0.33
**(B)**											
**Analyte**	**Puree (1)**		**Puree (2)**		**Puree (3)**			**Puree (4)**		**LOD**	**LOQ**
BPE	1.37		0.56		0.67			0.53		0.25	0.84
BPA	<LOQ		<LOD		12.72			<LOQ		0.51	1.71
BPB	<LOD		<LOD		<LOD			1.37		0.10	0.33
BPS	6.65		8.12		11.11			<LOD		1.66	5.55
**(C)**											
**Analyte**	**Rav. (** **1)**	**Rav. (2)**	**Rav. (3)**	**Rav. (4)**	**Rav. (5)**		**Far. (1)**	**Far. (2)**	**Tuna**	**LOD**	**LOQ**
BPAF	<LOD	<LOQ	<LOQ	<LOQ	<LOQ		<LOQ	<LOQ	<LOD	0.39	1.32
BPE	<LOD	<LOD	<LOD	<LOD	<LOD		<LOD	<LOD	1.28	0.25	0.84
BPA	21.67	26.44	36.80	26.18	22.13		40.65	37.47	10.82	0.51	1.71
BPB	1.12	1.79	3.91	1.92	1.83		6.90	5.11	<LOD	0.10	0.33
BPS	7.42	7.13	6.74	10.44	5.58		<LOD	<LOD	6.55	1.66	5.55
BPM	<LOD	<LOD	<LOD	<LOD	<LOD		<LOD	<LOD	4.36	0.50	1.65
**(D)**											
**Analyte**	**Lemon**		**Cola light**		**Beer (1)**			**Beer (2)**		**LOD**	**LOQ**
BPA	<LOQ		<LOQ		<LOQ			<LOQ		0.51	1.71
TMBPF	<LOD		<LOD		5.62			1.02		0.27	0.92
BPS	<LOD		<LOD		<LOQ			<LOQ		1.66	5.55

**Table 6 toxics-11-00665-t006:** BPA results obtained from the LC-MS/MS analysis in comparison with the GC-MS ones. Concentrations expressed in μg/kg.

Samples	BPA (LC-MS/MS)	BPA (LC-MS/MS) Recovery (%)	BPA (GC-MS)
Ravioli (4)	26.84	67	26.18
Ravioli (5)	22.33	81	22.13
Farce vol-aux-vent (2)	27.35	92	37.47

## Data Availability

Not applicable.

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
