# Peer review of "Simultaneous Quantification of 16 Bisphenol Analogues in Food Matrices"

_toxics, 2023, doi:10.3390/toxics11080665_

Round 1

Reviewer 1 Report

The manuscript presented the simultaneous quantification of 16 bisphenol analogues in food matrices, it has some merits, some comments are listed as follows:

1.    As mentioned in the introduction, liquid chromatography or gas chromatography, combined with mass spectrometry (LC-MS and GC-MS) or tandem mass spectrometry (LC-MS/MS and GC-MS/MS) are commonly used to detect and quantify bisphenols in food matrices; these methods are only suitable for specific bisphenol analogues, so why not for a broad range of BPs?

2.    The authors mention in the introduction that European Union regulations have focused on detecting the limits for BPA products across a range of sectors. Please specify the limit.

3.    Lines 124-125 in Materials and Methods, why add 100 μL ISTD after taking a homogeneous or liquid sample?

4.    Where are the parameters such as retention times of the analytes enumerated in Tables 1 and 2 derived, please cite references.

5.    The format of the table in Table 5 is not consistent, please adjust it

6.    What are the advantages of QuEChERS GC-MS compared with QuEChERS LC-MS/MS analysis?

7.    Please check the reference format for consistency, such as journal name, punctuation.

Reviewer 2 Report

In this manuscript, the authors investigated the effectiveness of the new method, QuEChERS (Quick, Easy, Cheap, Effective, Rugged, and Safe) GC-MS they developed to quantify the concentration of Bisphenols (BPs) including Bisphenol A in canned foods and beverages. The authors described that the methods allowed for selective and efficient quantitative analysis of the BPs compared to QuEChERS LC-MS/MS method and showed the presence of BPs in 20 of the 22 samples tested using QuEChERS GC-MS. This manuscript is important and helpful to analyze the presence and concentration of BPs quantitively. Before I recommend this manuscript is acceptable for publication, this manuscript should be revised about some points listed below.

1. In the introduction and discussion sections, the authors should introduce the other methods including the extraction for analyzing BP analogues reported by the other researchers. Then, the authors describe and discuss which points of the method are superior to the others.

2. Is there any information on the BPs in canned foods and beverages from the suppliers of cans and/or foods? I am wondering whether the BP analogues are from cans or foods or both. The authors should describe the BP origins.

3. In Material and Methods, section 2.2., the authors used Evian water to facilitate liquid-liquid extraction. I think that ultra-pure water, or Milli-Q water, should be used because Evian water may contain any BPs. Please explain why you used the water.
